# Wide-range and area-selective threshold voltage tunability in ultrathin indium oxide transistors

Robert Tseng[1], Sung-Tsun Wang[1], Tanveer Ahmed[1], Yi-Yu Pan[1], Shih-Chieh Chen[1], Che-Chi Shih[2], Wu-Wei Tsai[2], Hai-Ching Chen[2], Chi-Chung Kei[3], Tsung-Te Chou[3], Wen-Ching Hung[4,5], Jyh-Chen Chen[4], Yi-Hou Kuo[6], Chun-Liang Lin [6], Wei-Yen Woon[2] ✉, Szuya Sandy Liao[2] & Der-Hsien Lien [1] ✉

The scaling of transistors with thinner channel thicknesses has led to a surge in research on two-dimensional (2D) and quasi-2D semiconductors. However, modulating the threshold voltage ($V_T$) in ultrathin transistors is challenging, as traditional doping methods are not readily applicable. In this work, we introduce a optical-thermal method, combining ultraviolet (UV) illumination and oxygen annealing, to achieve broad-range $V_T$ tunability in ultrathin $In_2O_3$. This method can achieve both positive and negative $V_T$ tuning and is reversible. The modulation of sheet carrier density, which corresponds to $V_T$ shift, is comparable to that obtained using other doping and capacitive charging techniques in other ultrathin transistors, including 2D semiconductors. With the controllability of $V_T$, we successfully demonstrate the realization of depletion-load inverter and multi-state logic devices, as well as wafer-scale $V_T$ modulation via an automated laser system, showcasing its potential for low-power circuit design and non-von Neumann computing applications.

The field-effect transistor (FET) is a fundamental component in modern electronics. The threshold voltage ($V_T$) of a FET is a key parameter that determines its electronic functions, not only defining its switching modes but also scaling the supply voltage in the logic gates. In modern circuits, transistors are often tailored with different $V_T$ during the manufacturing process to enhance performance and lower power consumption. The ability to precisely control the $V_T$ of a transistor allows it to be used as a memory element, such as flash memory. In recent years, beyond-binary tunability between the "1" and "0" states has become particularly interesting in post–von Neumann applications, such as neuromorphic computation[1,2], multi-state memory[3,4] and multiplexed sensing[5]. The capacity to finely tune $V_T$ over a wide range is crucial for these applications, as it directly impacts the storage capacity and weight precision of associated devices. It enables the

development of advanced technologies that can perform complex tasks and have a wide range of applications in fields such as artificial intelligence, machine learning, and the Internet of Things (IoT).

Atomically thin semiconductors, including two-dimensional and other quasi-2D materials, have shown great potential to develop beyond-silicon electronics. Indium oxide ($In_2O_3$) recently emerged as a promising channel material for FETs as it can be thinned down to 1 nm and maintains high electron mobility beyond 100 $cm^2$ $V^{-1}$ $s^{-1}$, showing high on-state drain current density $I_D > 20$ A $mm^{-1}$ (refs. 6,7). This advancement allows for the scaling of $In_2O_3$ to align with modern technology nodes, and is expected to complement silicon-based systems for future back-end-of-line (BEOL) integration, expanding its range of applications beyond display technology. Moreover, $In_2O_3$ exhibits high potential to engineer the $V_T$ as a benefit generally granted

[1]Institute of Electronics, National Yang Ming Chiao Tung University, Hsinchu, Taiwan. [2]Research & Development, Taiwan Semiconductor Manufacturing Company, Hsinchu, Taiwan. [3]Taiwan Instrument Research Institute, National Applied Research Laboratories, Hsinchu, Taiwan. [4]Department of Mechanical Engineering, National Central University, Jhongli City, Taiwan. [5]K-Jet Laser Tek Inc., Hsinchu, Taiwan. [6]Department of Electrophysics, National Yang Ming Chiao Tung University, Hsinchu, Taiwan. ✉ e-mail: wywoona@tsmc.com; dhlien@nycu.edu.tw

to the oxide semiconductors (OS)[8]. In modern FETs, $V_T$ is tuned during fabrication by adjusting the implanted dopant concentration or by modifying the work function of the gate. To enable dynamic control over $V_T$, additional charge-trapped layers are required, e.g., a floating gate[9,10]. In OS-based FETs, $V_T$ can be tuned by various approaches such as post-fabrication thermal annealing[11,12], chemical doping[13], passivation[14], metal decorations[15], and incorporating a gate electrode with selected work functions[16], etc. It is known that vacancy-related surface effects play a crucial role in altering the charge carrier density of the OS, leading to the ease of $V_T$ tunability. Even though the vacancy-related surface effects can be easily controlled, scalable $V_T$ modulation with fine control over wide ranges is still challenging.

In this work, we report wide-range $V_T$ tunability in ultrathin $In_2O_3$ FETs achieved via a simple optical-thermal combined method. The method involves alternating ultraviolet (UV) illumination and oxygen annealing to achieve negative and positive $V_T$ tuning, respectively. The $V_T$ of a 2 nm-thick $In_2O_3$ transistor (30 nm-thick $SiO_2$ as the dielectric) exhibits a tunable window of 20 V with a resolution of 0.05 V, equivalent to a maximum change of sheet carrier density ($n_{2D}$) from $2 \times 10^{10}$ cm$^{-2}$ to $2 \times 10^{12}$ cm$^{-2}$ with a resolution of $10^9$ cm$^{-2}$. Importantly, this method is an entire post-fabrication process, and the $V_T$ modulation is reversible. We show that the $V_T$ of distributed transistors in a circuit can be selectively tuned, enhancing the gain of a depletion-load inverter by an order of magnitude. Exploiting the spatial $V_T$ tunability, we show that such local control of $V_T$ can be used to pattern the $V_T$ profile in a channel of a transistor to enable multi-step transfer characteristics, advocating potential for innovative logic design and neuromorphic applications. To further demonstrate the wafer-scale practicality, we fabricated $In_2O_3$ transistors on a 4-inch wafer and utilize the industry-level laser illumination system to achieve automatic large-area $V_T$ tuning for selected $In_2O_3$ transistors across the entire wafer.

## Results and discussion

Indium has been utilized across a variety of applications (element scarcity is shown in Supplementary Table 1), including displays (e.g., IGZO), transparent electrodes (e.g., ITO), and solar cells (e.g., CIGS). In this study, ultrathin $In_2O_3$ is employed as the channel material of a transistor, as shown schematically in Fig. 1a. The transistor is fabricated using atomic layer deposition (ALD) to form a 2–4 nm $In_2O_3$ layer on p$^{++}$Si substrates with a 30 nm-thick $SiO_2$. Nickel is used as the source and drain contacts. It is important to note that the entire fabrication process is compatible with back-end-of-line (BEOL) CMOS processes, as it is carried out below the thermal budget of 300 °C ("Methods")[17]. The

high-resolution transmission electron microscopy (HRTEM; Supplementary Fig. 1) image reveals the amorphous nature and the atomic level uniformity of the $In_2O_3$ films is confirmed by AFM (Supplementary Fig. 2), which effectively accounts for the minimal device-to-device variation (Supplementary Fig. 3). It has been shown that $In_2O_3$ shows high electron mobilities even in an amorphous phase[18]. This is because the electronic property of OS is insensitive to crystallinity. Their transport is determined by the overlapping of s orbitals between neighboring metal atoms (e.g., indium) and is less affected by structural disorders[8]. The Tauc plot extracted from absorption spectra (Fig. 1b and Supplementary Fig. 4) shows that the bandgap of $In_2O_3$ increases with decreasing channel thickness ($t_{ch}$), from 2.7 eV ($t_{ch} = 4$ nm) to 3 eV ($t_{ch} = 2$ nm) due to the enhanced quantum confinement effect. This finding is similar to the layer-dependent bandgap changes observed in 2D materials, revealing the quasi-2D feature of the ultrathin $In_2O_3$ films[19].

Figure 1c shows the transfer characteristics ($I_D$–$V_G$) of 2 nm $In_2O_3$ transistors annealed at 200 °C in $O_2$ and $N_2$ environments. The transfer characteristics ($I_D$–$V_G$) of $In_2O_3$ below 2 nm is shown in Supplementary Fig. 5 and saturation behavior ($I_D$–$V_D$) is shown in Supplementary Fig. 6. $O_2$ and $N_2$ annealing lead to $V_T$ shifts to +6 V and −20 V, respectively; both show effective tuning of $V_T$ after 30 min of annealing (the $V_T$ extraction method is described in the "Methods" section). While the traditional annealing method can effectively tune the $V_T$ of $In_2O_3$ transistors, precise control of $V_T$ over the tuning range is difficult. Figure 2a illustrates the proposed approach to achieve fine-tuning of $V_T$ in ultrathin $In_2O_3$ transistors. The device is first subjected to $O_2$ annealing, which shifts the $V_T$ to +6 V. Subsequent UV irradiation (365 nm) on $In_2O_3$ transistors shifts the $V_T$ to the negative values (Fig. 2b), where the magnitude of $V_T$ shifts depends on the exposure time, incident power density and wavelength (Fig. 2c and Supplementary Figs. 7 and 8). Note that UV exposure causes an accumulation of $V_T$ shift, leading to a saturation point of −15 V with a tuning resolution of 0.05 V within a tunable window $\Delta V_T$ of 21 V (Supplementary Fig. 9). To verify reproducibility of our approach, we tested devices with identical fabrication processes but different metal contacts (Pd, Pt) and substrates (10 nm $HfO_2$ as the gate dielectric). All results exhibited consistent trends, regardless of substrate and metal contacts. (Supplementary Fig. 10). Importantly, the $V_T$ modulation using the proposed approach is reversible. UV-exposed $In_2O_3$ transistors with negative $V_T$ can be reset to have a value of 5 V after $O_2$ annealing. Figure 3a shows repeated $V_T$ modulation achieved by multiple $O_2$ annealing-UV exposure cycles. The field effect mobility ($\mu_{FE}$) of the

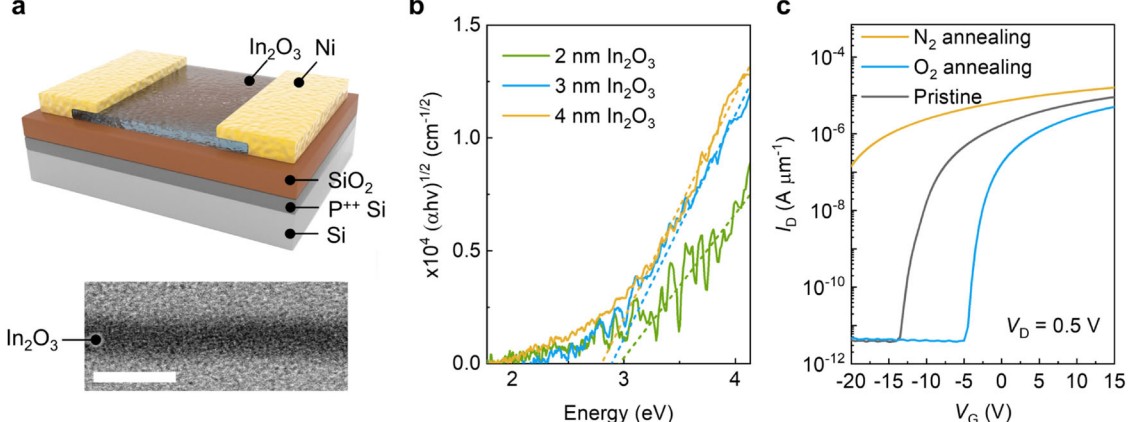

**Fig. 1 | Device structures and high-resolution transmission electron microscopy (HRTEM) images of atomic layer deposition (ALD) deposited $In_2O_3$ transistors. a** Schematic of a 2 nm $In_2O_3$ transistor. The inset shows the HRTEM image of the ALD-deposited ultrathin $In_2O_3$ films. The scale bar is 5 nm. **b** Tauc plot of the $In_2O_3$ films as a function of film thickness. The fitting of the curve is done

based on the Tauc model (α: absorption coefficient; h: Planck's constant; v: frequency of vibration). **c** Transfer curves of 2 nm $In_2O_3$ transistors with channel width/length of 10/2 μm and annealed under $N_2$ and $O_2$ for 30 min at 150 °C. Note that the amounts of threshold voltage ($V_T$) shifts are saturated after 30 minutes ($I_D$: drain current; $V_D$: drain voltage; $V_G$: gate voltage).

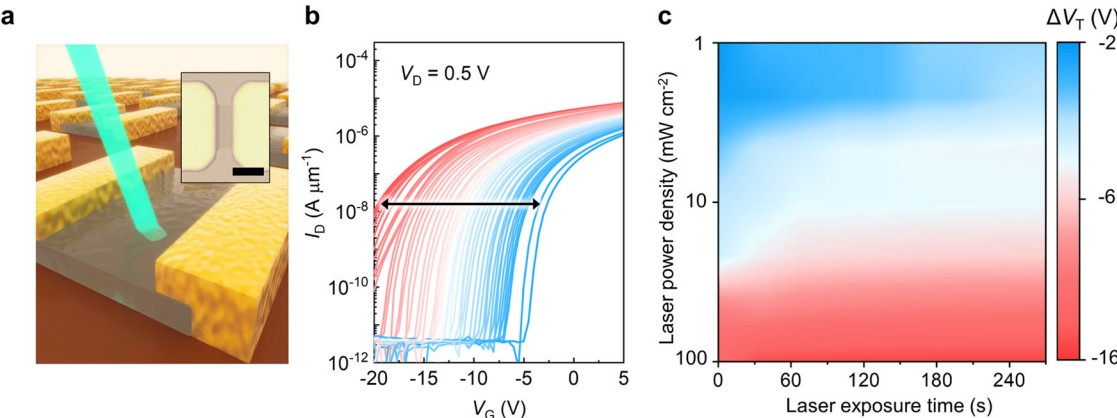

**Fig. 2 | $V_T$ tuning in ultrathin $In_2O_3$ transistors via ultraviolet (UV) exposure. a** A schematic of $V_T$ tuning in ultrathin $In_2O_3$ transistors through UV exposure combined with thermal annealing; the inset is the microscopic image of the transistor with a scale bar of 5 μm. **b** Transfer curves of 2 nm $In_2O_3$ transistors with channel width/length of 10/2 μm after post treatments. The arrow represents the transition of transfer curves with UV light exposure (red lines) and $O_2$ annealing (blue lines) ($I_D$: drain current; $V_D$: drain voltage; $V_G$: gate voltage). **c** A contour plot of $V_T$ shifts as a function of UV exposure time and power density. Devices were annealed at 150 °C in $O_2$ for 30 min to reset the $V_T$ before each UV exposure measurement. The absorbed power density increasing from $1 \times 10^6$ mW cm$^{-2}$ to $1 \times 10^8$ mW cm$^{-2}$ for exposure times from 30 s to 300 s under 365 nm laser illumination. The measurement time interval is 30 s and the transfer characteristics of the device are immediately measured (<5 s) after UV illumination. The plot consists of 7 different power densities and 10 different exposure times.

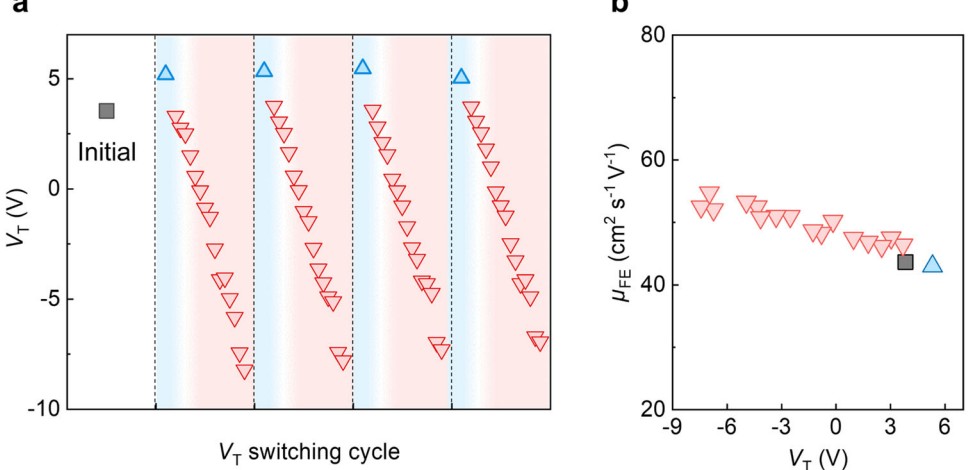

**Fig. 3 | Reversibility of $V_T$ tuning in ultrathin $In_2O_3$ transistors. a** $V_T$ of a 2 nm $In_2O_3$ transistor with channel width/length of 10/2 μm during multiple $O_2$ annealing and UV exposure cycles. $In_2O_3$ transistors are thermally annealed under $O_2$ for 30 min, followed by UV illumination under a power density of 1 mW cm$^{-2}$. The blue triangles are the $V_T$ after $O_2$ annealing. The red triangles are the $V_T$ after UV exposure. The dashed lines separate different switching cycle and the shaded areas represent the treatments (blue: $O_2$ annealing; red: UV exposure). **b** The field effect mobility ($\mu_{FE}$) of ultrathin $In_2O_3$ transistors as a function of $V_T$ tuned by the proposed method.

transistors is maintained at $49 \pm 5$ cm$^2$ V$^{-1}$ s$^{-1}$, indicating that the properties of the $In_2O_3$ transistors are not significantly affected by the tuning $V_T$ processes (Fig. 3b). Additionally, we also conduct the bias stress experiment to investigate behavior and reliability of transistors under different bias conditions after the treatment of $O_2$ annealing and UV exposure (Supplementary Fig. 11). A gate voltage was applied at ±15 V for 1000 s, with both source and drain grounded. The transfer characteristics of the device are immediately measured (<1 s) after bias stress. Furthermore, to evaluate the potential damage caused by laser exposure, we performed finite element simulations to analyze the temperature changes. The results indicate only a slight increase in temperature at the applied incident power densities (Supplementary Fig. 12).

The size of the tunable window $\Delta V_T$ is dependent on the thickness of the $In_2O_3$, as shown in Fig. 4a, b ($V_T$ shift as a function of channel length is shown in Supplementary Fig. 13). $In_2O_3$ with a thickness of 4 nm shows the largest tunable window, whereas $In_2O_3$ with a thickness

of 2 nm is capable of being tuned between the enhancement mode ($V_T > 0$ V) and the depletion mode ($V_T < 0$ V). The offsets of tunable window for different thickness can be explained by the quantum confinement effect on the trap-neutral level model[19]. Fermi level is located deeply inside the conduction band for thicker $In_2O_3$ and aligns within the bandgap for thinner $In_2O_3$ as the bandgap is enlarged due to the enhanced quantum confinement effect.

To further quantify the $V_T$ shift, we extract 2D carrier density ($n_{2D}$) using the Drude model $n_{2D} = I_D L / (qWV_D\mu_{FE})$, where $q$ is the electron charge, $I_D$ is the source-drain current at zero gate voltage, $V_D$ is the source-drain voltage, and $\mu$ is the field effect carrier mobility. Based on the Drude model, we benchmark our method against other $V_T$ tuning methods used in different classes of ultrathin semiconductors. Figure 4c shows the $n_{2D}$ of 2D semiconductors modulated by various chemical doping schemes[20–30], which ranges from $10^{12}$ to $10^{13}$ cm$^{-2}$ depending on the doping methods and materials (details in Supplementary Table 2). The tunable ranges of $n_{2D}$ achieved in $In_2O_3$

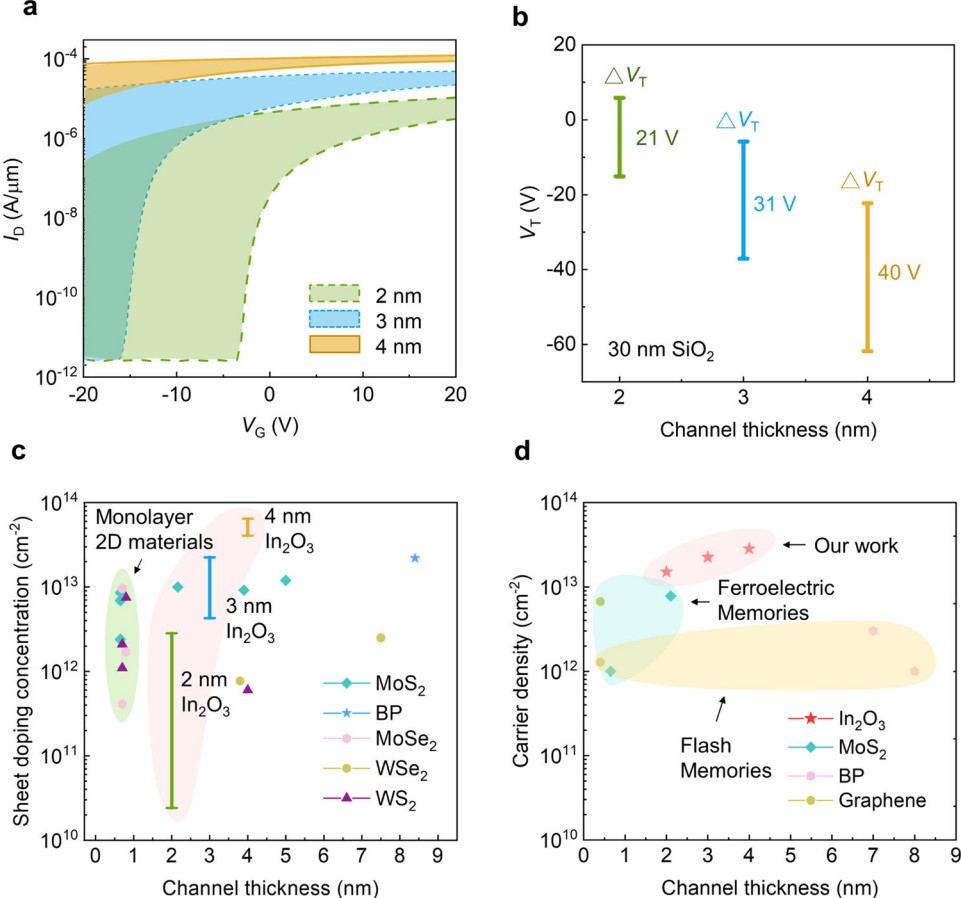

**Fig. 4 | Tunable $V_T$ windows and benchmarks of In$_2$O$_3$ transistors versus other ultrathin transistors. a** The range of the tunable window of In$_2$O$_3$ transistors with various In$_2$O$_3$ thicknesses, channel width/length of 10/2 μm. **b** The range of $V_T$ after treatment. **c** Benchmark of sheet carrier density ($n_{2D}$) for different classes of ultrathin semiconductors with different thicknesses[20–30], calculated using both the Drude model and the parallel-plate capacitor model. The shaded areas represent different classes of materials (red: In$_2$O$_3$; purple: monolayer 2D materials). **d** The $\Delta n_{2D}$ for different classes of ultrathin semiconductors based on capacitively charged structures. Shaded area represents different material categories (red: In$_2$O$_3$; green: ferroelectric memories; yellow: flash memories).

transistors vary from $10^{10}$ cm$^{-2}$ to $10^{13}$ cm$^{-2}$ depending on the thickness. 2 nm In$_2$O$_3$ exhibits the widest $n_{2D}$ tunable window ($2 \times 10^{10}$ cm$^{-2}$ to $2 \times 10^{12}$ cm$^{-2}$), indicating the proposed method is competitive among various 2D doping techniques. The proposed methods offer a comparable effect to chemical doping, while providing the advantages of reversibility and area selectivity. In contrast to the substantial challenges associated with chemical doping in 2D materials, our method showcases the ease of achieving effective carrier concentration tuning in ultrathin In$_2$O$_3$. Moreover, our approach minimizes fabrication processes, thereby enhancing its potential for seamless integration in BEOL applications.

Another commonly used approach to control the $V_T$ of a transistor is through device design, usually done by inserting a layer that can be capacitively charged. For instance, flash memories store charge in a charge-trapping layer to modulate $V_T$ through the trapping and de-trapping process. Similarly, ferroelectrics modulate $V_T$ by controlling the polarization of a ferroelectric layer. The density of stored charge carriers $n_{2D}$ is around $10^{12}$ cm$^{-2}$ to $10^{13}$ cm$^{-2}$ for graphene and related 2D materials-based flash memories and ferroelectric memories. The $\Delta n_{2D}$ for In$_2$O$_3$ extracted by the parallel-plate capacitor model ($1.5 \times 10^{13}$, $2.3 \times 10^{13}$ and $2.8 \times 10^{13}$ cm$^{-2}$ for 2, 3 and 4 nm, respectively) is of the same order as flash memories and ferroelectric memories without extra gates (Fig. 4d)[9,31–35]. The details of 2D-based charge-storing schemes are summarized in Supplementary Table 3. Compared to other 2D doping and capacitively charged techniques, the proposed method demonstrates a more effective way to largely tune the carrier

density in In$_2$O$_3$ transistors, enabling In$_2$O$_3$ transistors for innovative circuit and memory applications.

We conducted cyclically annealing between N$_2$ and O$_2$ environments to investigate the mechanism (Supplementary Fig. 14). The results demonstrate that the $V_T$ shift by N$_2$ and O$_2$ annealing is reversible, similar to the reversibility observed in UV exposure and O$_2$ annealing processes. This result implies that the mechanisms underlying these two approaches may be identical. It is known that electronic properties of amorphous In$_2$O$_3$ can be altered by oxygen-related defects such as oxygen vacancies and oxygen adatoms on the surface[36]. Oxygen vacancies act as shallow donors and contribute to the spontaneous n-type conductivity of the In$_2$O$_3$ (the shallow donor level is observed in scanning tunneling microscopy (STS) shown in Supplementary Fig. 15). Oxygen adatoms, on the other hand, act as acceptor-like traps that counter-dope the n-type OS. Annealing in oxygen-rich or oxygen-scarce (N$_2$ or vacuum) environments causes a rebalance of physically adsorbed oxygen adatoms, leading to positive and negative $V_T$ shifts, respectively. Furthermore, exposure to UV light generates holes that neutralize the negatively charged oxygen adatoms, resulting in their detachment and a decrease in the $V_T$. This proposed mechanism is supported by the measurement of $V_T$ retention in different atmospheres, as shown in Supplementary Fig. 16. Additionally, the $V_T$ exhibits almost no change over time in an ultra-high vacuum environment ($10^{-9}$ torr), which further highlights the crucial role of oxygen adatoms in this process. The results suggest that the stability could be improved by implementing effective

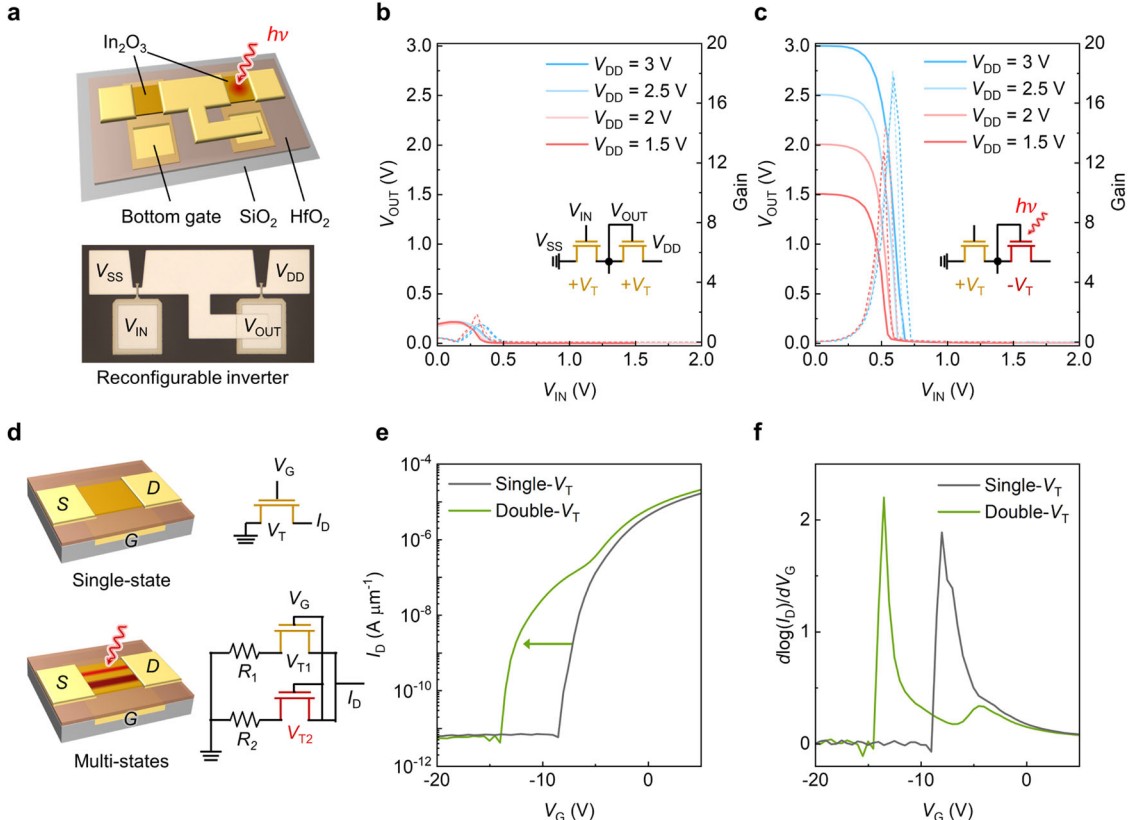

**Fig. 5 | Applications of ultrathin In$_2$O$_3$ transistor with laser exposure. a** A schematic and optical microscope image of In$_2$O$_3$ inverter (*h*: Planck's constant; *v*: frequency of vibration; $V_{DD}$: drain voltage; $V_{SS}$: source voltage; $V_{IN}$: input voltage; $V_{OUT}$: output voltage). **b** The voltage transfer characteristics of 2 nm In$_2$O$_3$ inverter with channel width/length of 10/2 μm before UV laser illumination. **c** The voltage transfer characteristics of In$_2$O$_3$ inverter of 2 nm In$_2$O$_3$ inverter with channel width/ length of 10/2 μm after UV laser illumination. **d** A schematic of ultrathin In$_2$O$_3$ transistor with multi-state logic function ($V_G$: gate voltage; $I_D$: drain current; $R_1$: resistance of first channel; $R_2$: resistance of second channel; $V_{T1}$: threshold voltage of first channel; $V_{T2}$: threshold voltage of second channel). **e**, The transfer characteristics of In$_2$O$_3$ transistor with UV laser illumination on the specific area. **f** The transfer characteristics with the first-order derivative of log($I_D$) from **e**.

---

isolation or passivation to minimize the exposure of In$_2$O$_3$ to the atmosphere.

## Demonstration of depletion-load inverter and multi-step logic

The proposed method allows for precise control of $V_T$, making it useful for advanced low-power circuits[37]. Here, we demonstrate a depletion-load inverter by adjusting the $V_T$ of the selected In$_2$O$_3$ transistors in a circuit, achieved by employing a micro-laser system that can focus UV exposure selectively on In$_2$O$_3$ transistors (details in "Methods"). Figure 5a illustrates the layout of an inverter circuit with two n-type In$_2$O$_3$ transistors connected by local bottom gates. Both transistors are set to enhancement mode via O$_2$ annealing, resulting in a $V_T$ shift to +5 V. When $V_{DD}$ is applied, $V_{OUT}$ is limited as the channel resistance of the load transistor is large, resulting in a small gain of 2 for $V_{DD}$ = 3 V (Fig. 5b). After exposing the load transistor to UV light, tuning the selected transistor to depletion mode, the inverter becomes a depletion-load inverter with an enhanced gain of 18 (Fig. 5c). Note that the depletion load inverter serves as a proof of concept for our proposed technology. Integrating a complementary MOS configuration has the potential to further optimize the performance of inverters, resulting in reduced power consumption and improved spatial advantages. The functionality of the circuit is activated via the proposed method with local $V_T$ tunability, giving a higher degree of freedom to design and calibrate the circuit even after fabrication.

The ability to locally adjust the $V_T$ with high spatial resolution in In$_2$O$_3$ transistors allows for creating non-uniform $V_T$ patterns in the channel and enables new functions. We demonstrate that a reconfigurable In$_2$O$_3$ transistor can switch between binary and multi-state logic without the need for multiple gates[38] or heterojunctions[39]. Figure 5d illustrates a transistor with uniform $V_T$ along the channel detailed in Supplementary Fig. 17. After UV exposure on the selected area, non-uniform $V_T$ pattern forms on the channel. The equivalent circuit configuration changes depending on the scanning pathways, consisting of parallel resistors and transistors with different $V_T$ depending on the exposure time. As a result, a single swing in transfer characteristics turns to double-swing characteristics (Fig. 5e, f), increasing the information capacity of a single transistor. The demonstration of an In$_2$O$_3$-based multi-state logic device introduces an alternative strategy to design reconfigurable multifunctional logic circuits and has the potential for neuromorphic applications (Supplementary Fig. 18)[40,41].

## Wafer-scale $V_T$ modulation

To further demonstrate the practicability of our technique, we fabricated In$_2$O$_3$ transistors on a 4-inch wafer (Fig. 6a) and employed an automated laser illumination system (Laser Lift-Off System; K-JET LASER TEK Inc.) for large-scale $V_T$ tuning (Fig. 6b). The system allows for precise alignment to direct focused flat-top illumination onto selected devices, enabling localized $V_T$ modulation (Fig. 6c, Supplementary Movie 1 and tool details in Methods). We demonstrate the scalability by tuning $V_T$ for 450 transistors in an 18 × 25 array (Fig. 6d). The initial $V_T$ was set to -7 V through O$_2$ annealing for all transistors (Fig. 6e), and the $V_T$ of the laser-exposed transistors were tuned to negative values (Fig. 6f). The transfer curves after $V_T$ modulation were shown in Fig. 6g, demonstrating the consistency of the proposed

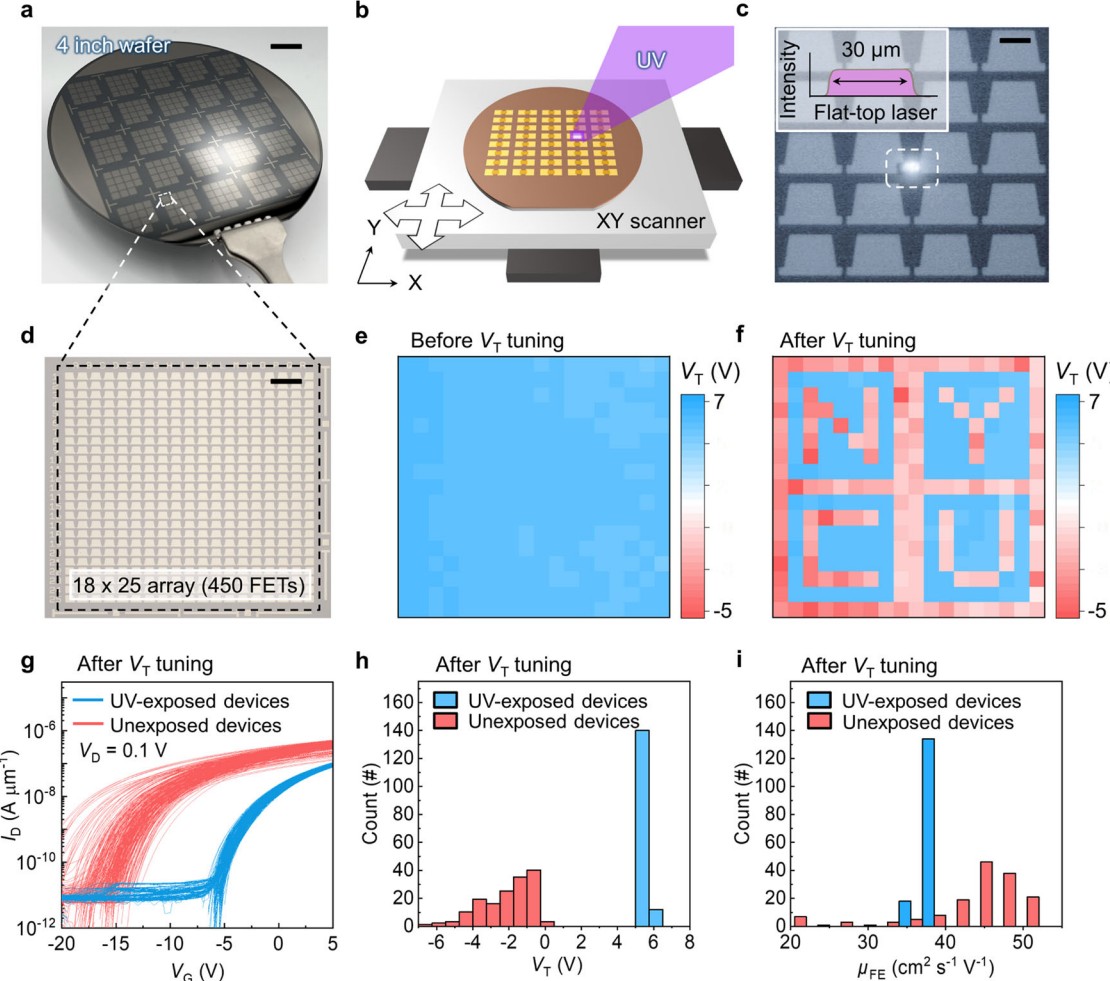

**Fig. 6 | Large-area $V_T$ modulation. a** An image of ultrathin $In_2O_3$ transistor arrays on a 4-inch wafer. The scale bar is 1 cm. **b** Schematics of the automated laser illumination system used for $V_T$ modulation. **c** An optical microscope image of UV laser exposure on the designated channel area. The inset is the illustration of flat-top laser beams. The scale bar is 50 μm. **d** An optical microscope image of the basic array unit on 4-inch wafer consisted of an 18 × 25 array (450 transistors). The scale bar is 200 μm. **e** A contour plot showing the $V_T$ of the transistors in the array before $V_T$ tuning. **f** A contour plot showing the $V_T$ of the transistors in the array after $V_T$ tuning. **g** The transfer curves of the devices with channel width/length of 10/2 μm in the array after $V_T$ tuning (red line: UV exposed areas; blue line: unexposed area). **h** The histogram of $V_T$ of the devices in the array after $V_T$ tuning. **i** The histogram of $\mu_{FE}$ of the devices in the array after $V_T$ tuning.

approach. The $V_T$ of unexposed transistors remained the same after the large-area tuning, as shown in Fig. 6h. The $\mu_{FE}$ distribution shows that the mobility increases slightly after UV exposure (Fig. 6i), in agreement with previous results. Note that the device-to-device variation, induced by the variation in film quality across the wafer, may potentially be mitigated through the utilization of industrial-level ALD tools. Furthermore, the $V_T$ tuning precision could be improved by the incorporation of patterned photomasks. This demonstration not only shows capability of wafer-scale fabrication and $V_T$ tuning on multiple transistors, but also opens up possibilities for enhancing circuit functionality beyond $V_T$ adjustment.

In summary, we demonstrate the wide-range $V_T$ tunability in ultrathin $In_2O_3$ using an optical–thermal method. This approach is comparable to doping and capacitively charged techniques used in other ultrathin semiconductors and avoids the need for external dopants and complex processes. Our demonstration of a depletion-load inverter suggests that this $V_T$ tuning method has the potential to enable new functionalities in combinational logic circuits. Additionally, we have demonstrated a single device with multi-state logic function, which opens up new possibilities for the design of multifunctional logic circuits and non-von Neumann computing. Moreover, we conducted large-scale $V_T$ modulation using an industry-level tool to

highlight the practical applicability of this method alongside individual device testing.

## Methods

### Device fabrication
The device fabrication started with a standard wet and dry pre-clean of 300 mm Si substrate. The highly phosphorus-doped Si, which carrier density is over $10^{21} cm^{-3}$, was grown on Si substrate as a global back-gate. Then, 30 nm $SiO_2$ was deposited by ALD at 260 °C with $(H_2Si[N(C_2H_5)_2]_2)$ as the precursor. Afterwards, $In_2O_3$ thin films with different thicknesses were deposited by ALD at 200 °C using $(CH_3)_3In$ (TMIn) and $O_3$ as indium (In) and oxygen (O) precursors. The active areas of $In_2O_3$ were defined by lithography with HCl etching with the channel width/length of 10/2 μm. Standard lithography patterning and lift-off procedures were performed to contact the $In_2O_3$ thin films with metal electrodes. 40 nm Ni was deposited by e-beam evaporation on $In_2O_3$ to serve as source/drain ohmic contacts.

### Inverter fabrication
The inverter fabrication started with a standard wet and dry pre-clean of 250 mm Si with a 50 nm $SiO_2$ substrate. Then, the local back gate of

40 nm Ni was deposited by e-beam evaporation. Afterwards, 6 nm $HfO_2$ was deposited by ALD at 250 °C with TDMAHf as a precursor for 70 cycles. After the gate dielectric deposition, 2 nm $In_2O_3$ thin films were deposited by ALD at 200 °C using $(CH_3)_3In$ (TMIn) and $O_3$ as In and O precursors. The active areas of $In_2O_3$ were defined by lithography with HCl etching for 5 s. Then, the contact holes were patterned by standard lithography, followed by BOE etching for 5 min. Finally, a Ni (40 nm) film was deposited by e-beam evaporation to serve as source/drain ohmic contacts and metal line interconnections.

### Device characterization

As-grown $In_2O_3$ was then measured for thickness using a transmission electron microscope (TEM). A focused ion beam (FIB) system (Auriga, Carl Zeiss) was used to fabricate the cross-sectional specimen, which was then examined by a TEM (none-Cs Metrios). The electronic characteristics were measured by an Agilent B2902B source. The $\mu_{FE}$ and $V_T$ of ultrathin $In_2O_3$ transistors were determined in linear regime using conventional MOSFET equation for $V_D \ll V_G - V_T$: $I_D = \frac{W}{L} \mu_{FE} C_{OX} (V_G - V_T) V_D$, where $C_{OX}$ is the oxide capacitance. The result of $\mu_{FE}$ can be obtained from $\mu_{FE} = \frac{L g_m}{W C_{OX} V_D}$, where $g_m$ is maximum transconductance. The resulted $V_T$ was determined by linear extrapolation, which was conducted by plotting $I_D$ versus $V_G$, extrapolating from maximum transconductance ($g_m$) to $I_D = 0$ and adding $V_D/2$ to obtain the intercept at $V_G$ axis.

### Laser exposure

The 365 and 532 nm laser beams were generated by diode lasers (RGB Laser systems, Lambda Beam). The laser light passes through a shutter (NM Laser Product, LST-5VDC) to control the laser exposure time. Following the shutter, an ND filter was used to control the laser incident power. After that, the laser beam (365, 532 nm) was focused on the device through a fixed optical path using a ×50 objective lens (OLYMPUS, LMPlanFL N ×50, NA = 0.5) with a laser spot diameter of ~10 μm. Meanwhile, the absorbed power was measured by a power meter which comprised a detector (Thorlab, S120VC) and a meter (Thorlab, PM100D).

### Thermal annealing

The fabricated devices were placed in a customized chamber with two gas inlets. Then, the devices were annealed in $O_2$ and $N_2$ with 1 liter per minute gas flow for 30 min at 150 °C. The pressure was kept at about 1 atm.

### Absorption spectra

The $In_2O_3$ films were deposited on a 200 μm-thick glass substrate following the same ALD process as mentioned before. An ellipsometer determined the thickness of $In_2O_3$ films for absorption measurement. A Hitachi U-4100 spectrophotometer was used to investigate the transmittance of samples between the 300 and 700 nm range with an integrating sphere. Absorptance of the films was calculated using the relation $A = 100 - (T + R)$.

### Wafer-scale $V_T$ modulation

Wafer-scale $V_T$ modulation was achieved using an automatic laser illumination tool (Laser Lift-Off System; K-JET LASER TEK Inc.). The laser spot size and shape can be customized by a mask based on our specific needs, allowing for arbitrary adjustments. The laser intensity is uniformly distributed across the entire exposure area (20 μm × 1 μm for this study), commonly known as flat-top beams, achieved via applying a mask on the optical path to ensure consistent laser exposure within the targeted region. Laser pulses with energy of 0.0015 J/cm², a wavelength of 365 nm, and a pulse width of 26 ns were directed onto the desired regions or devices. The scanning rate is ~1 ms/transistor.

## Data availability

Relevant data supporting the key findings of this study are available within the article and the Supplementary Information file. All raw data generated during the current study are available from the corresponding authors upon request.

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

## Acknowledgements

This work was supported by the National Science and Technology Council, Taiwan, under Grant No. NSTC 110-2112-M-009-010-MY3. This work was in part supported by the "Advanced Semiconductor Technology Research Center" from The Featured Areas Research Center Program within the frame work of the Higher Education Sprout Project by the Ministry of Education (MOE) in Taiwan and in part under Grant No. NSTC 111-2634-F-A49-008. This work was in part supported by Semiconductor Research Corporation under Grant No. SRC 2022-PK-3801. The research was supported by Taiwan Semiconductor Manufacturing Company. D.-H.L. acknowledges the Yushan Scholar Program by the MOE in Taiwan. R.T. would like to thank Dr. Hsin-Fu Kuo from GoMore for scholarship financial support. The author also thanks Dr. Yu Ming Lin and Dr. Chung Te Lin for resource support in sample preparation.

## Author contributions

R.T. and D.-H.L. conceived the idea and designed the experiment. W.-Y.W. and D.-H.L. provided scientific guidance throughout. R.T., S.-T.W., Y.-Y.P., and S.-C.C. performed the device fabrication and the measurement. C.-C.K. and T.T.C. designed and developed the ALD tools. W.-Y.W. and S.S.L. provided engineering guidance for ALD growth. C.-C.S., W.-W.T. and H.-C.C. performed the ALD growth and TEM. Y.-H.K. and C.-L.L. performed the STS. W.-C.H. and J.-C.C. designed the laser illumination system and performed large-area VT tuning experiments. R.T., S.-T.W., W.-Y.W. and D.-H.L. analyzed the data. R.T., T.A., and D.-H.L. wrote the paper. All authors have read and approved the final version of the paper.

## Competing interests

The authors declare no competing interests.
