## [Peer review file · Nature Communications]

REVIEWER COMMENTS

Reviewer #1 (Remarks to the Author):

This work demonstrates an effective VT tuning method by UV laser for negative shift and O2 for positive Vt shift. However, the proposed method seems not useful for large-area fabrication. Please see below for my comments.

1. The UV laser spot size is ~ 10 μm . Can this method be modified for large area fabrication? The technology proposed in this work seems can only be used in a single device or "a few" devices.
2. Why N2 annealing lead to a negative shift, as show in Fig. 1? What is the difference between N2 + O2 annealing and UV laser + O2 annealing?
3. What is the impact of UV and O2 annealing on PBS and NBS performance?
4. The benchmark in Fig. 4b is not fair. For FE memory or Flash memory, Vt shift is controlled by voltages. However, Vt shift in this work is by UV laser. This is no way to use UV laser to control Vt in memory applications.

Reviewer #2 (Remarks to the Author):

The manuscript "Wide-range and area-selective threshold voltage tunability in ultrathin indium oxide transistors" is an interesting report on threshold voltage tunable thin indium oxide transistors. The main noteworthy results shown regard the wide tunability of the threshold voltage in indium oxide TFTs usable for BEOL applications. The work is of significance to the field of transistors, both flexible and BEOL integrated, also in view of neuromorphic computing applications. It definitively show advancements as compared to the state of the art of the field, even tough the use of thin and very conductive indium oxide layers as semiconductors is opinable. The conclusions are not sufficiently supported, and several major questions are still open (as detailed below). The methodology used is sufficiently sound, even if more details in the methods are required (see details below).

The following aspects require a major revision, adressing properly each of them:

- 1) In the introduction, authors mention an off current (I_{off}) of 20A/mm, which is surely not a current but a current density. Please correct this. Also please clearly talk about I_{off} .
- 2) Besides desirable and induced Vth shifts, authors should clearly discuss and present further Vth variations, e.g., with bias stress, light, aging, mechanical forces, etc.
- 3) there are several typos like "atomic layer disposition (ALD)". Please correct them.
- 4) Authors should clearly discuss about the disadvantages of using depletion mode TFTs, especially in terms of power consumption and circuit design. Is there any possibility to have a positive Vth and how?
- 5) authors only show linear curves. What is the saturation behavior of the TFTs? What higher voltages can be used with such a thin OS?
- 6) authors should be clearly discuss about the cost and practicability of the proposed technique, also presenting results of possible damages in terms of BEOL laser damage to the other devices/layers?
- 7) authors present mobility values, are those effective mobility values? how do these values change for different lengths?
- 8) authors do not show the performance of the devices with different W/L ratios? Actually W/L ratios are not clearly presented in the methods.
- 9) The inverters shown are still unipolar and not complementary. Authors should clearly discuss about pro/contra of this method as compared to complementary materials, also in terms of power consumption considering that the threshold voltage is negative.
- 10) Authors should discuss about availability of indium, when presenting the different semiconductor materials. A true comparison with

all relevant parameters is required.

11) authors use ALD deposited indium oxide? could also sputtering (mostly used in OS processing) be used?

12) could ALD be used even below 2 nm?

13) authors do not really comment on the semiconductivity of such a thin indium oxide layer. what about the saturation mobility?

14) authors do not show results with different channel dimensions, semiconductors island dimensions. A study of this would be really useful to be added.

15) authors conclude that this method is comparable to doping or capacitively charged techniques. it is indeed not clear in what this method is better, this is in fact very important.

Reviewer #3 (Remarks to the Author):

In this report, Tseng et. al. use a combination of thermal annealing and UV light exposure to tune the threshold voltage of transistors based on very thin (< 5nm) films of In₂O₃. These are interesting devices which are gaining more interest in the community, and the authors contribution provides useful insights into how one might be able to tune the threshold voltage of these transistors.

Unfortunately however, I cannot recommend this manuscript for publication in its current form. While the results are reasonable, they primarily describe a strategy to control devices in the group's own lab, rather than being of use to the general audience. To elaborate on this slightly, these are results for a specific device design, no device-to-device variation is provided, and the effect is parameterised in terms of voltage, which depends on the properties of the interface, device dimensions and so on. Overall, I don't disagree with the methods, motivation, nor the conclusions drawn; I just feel the contribution is not broad enough to warrant publication in Nature Communications unfortunately.

To be able to recommend this paper for publication in Nature Communications I would need to see:

1) Quantification of device-to-device variation with multiple, identically fabricated, devices on different substrates.

2) A description of how VT depends on geometry. I am confident that it would depend on both W and L.

3) Confirmation that this is not unique to the exact structure depicted in Figure 1a. It would not be necessary to consider every possibility of course. But it would need to be confirmed that the VT shifts were in the same directions if using, say, different electrode materials, for example.

4) There needs to be some comment on the stability / longevity of this strategy. E.g. if UV light is used to shift the threshold voltage, how long does the state remain for? I suspect these VT shifts are quasi-stable.

A few other more minor concerns:

5) The introduction reads as if In₂O₃ is competing with silicon. In₂O₃ is better suited for large-area applications like display backplanes and so-on than for information processing. I would consider re-writing this.

6) Can the authors comment on the continuity / roughness of the films. 2nm is very thin, and with ALD I would not be surprised if there were pinholes or significant thickness variability.

7) The following statement needs to be rephrased slightly. "This is because the electronic property of OS is insensitive to crystallinity. Their transport is determined by the overlapping of s orbitals between neighbouring metal atoms (e.g. indium) and is not affected by structural disorders." The effect of crystallinity on transport is certainly lower in metal oxides compared to covalent systems (e.g. a:Si-H) but order does still have some affect. I would use the phrase "less affected by structural disorders"

rather than “not affected by structural disorders”.

8) For the heatmaps (e.g. Figure 2(c), Figure S4), how many data points are used? This looks like it is interpolated / smoothed?

9) For Figure 2(c) (and similar measurements), what was the time between measurements and what were the conditions between measurements? I strongly suspect that illumination and annealing history would have a notable impact on these results.

10) I’m not sure Figure 2a really adds anything. It feels more like a Table of Contents graphic than a subfigure.

11) I think the units for mobility on page 5 should be cm^2/Vs not m^2/Vs .

Answers to Reviewers' Comments

We appreciate the reviewers for their time and insightful questions. Please find the responses to their comments below. Our responses to the reviewers' questions are shown below. Please note that the reviewers' comments are in black italic font and all changes made to the manuscript and Supplementary Information are highlighted in bold.

Reviewer #1 (Remarks to the Author):

This work demonstrates an effective V_T tuning method by UV laser for negative shift and O_2 for positive V_T shift. However, the proposed method seems not useful for large-area fabrication. Please see below for my comments.

1. The UV laser spot size is $\sim 10 \mu\text{m}$. Can this method modify for large area fabrication? The technology proposed in this work seems can only be used in a single device or “a few” devices.

We appreciate the comments provided by the reviewer. We fully acknowledge the importance of showcasing the capability of our technique for large-area fabrication in order to demonstrate its practicability. For that, we fabricated wafer-scale In_2O_3 transistors on a 4-inch wafer and utilize the industry-level tool (Laser Lift-Off System; K-JET LASER TEK Inc.) to achieve automatic large-area V_T tuning for selected In_2O_3 transistors across the entire wafer. This system (the Laser Lift-Off System; K-JET LASER TEK Inc.) was originally designed for μ -LED mass transfer but proved to be suitable for our purposes. The laser spot size and shape can be customized by a mask based on our specific needs, allowing for arbitrary adjustments. The laser intensity is uniformly distributed across the entire exposure area, commonly known as “flat-top” beams, which ensure consistent laser exposure within the targeted region. By utilizing this system, we were able to customize the V_T of selected devices in an 18×25 (450) FET array. This demonstration not only shows capability of wafer-scale fabrication and V_T tuning on multiple transistors but also opens up possibilities for enhancing circuit functionality beyond V_T adjustment. To provide a clear visual representation of our new results, we have included a new figure (Fig. 6) in the manuscript. Additionally, a new paragraph has been added to manuscript and details of the techniques are updated in the Method section. We also edited the abstract, introduction and conclusions accordingly.

(page 9, line 232)

Wafer-scale V_T modulation

To further demonstrate the practicability of our technique, we fabricated In_2O_3 transistors on a 4-inch wafer (Fig. 6a) and employed an automated laser illumination system (Laser Lift-Off System; K-JET LASER TEK Inc.) for large-scale V_T tuning (Fig. 6b). The system allows for precise alignment to direct focused flat-top illumination onto selected devices, enabling localized V_T modulation (Fig. 6c, Supplementary video 1 and tool details in Methods). We demonstrate the scalability by tuning V_T for 450 transistors in an 18×25 array (Fig. 6d). The initial V_T was set to $\sim 7 \text{ V}$ through O_2 annealing for all transistors (Fig. 6e), and the V_T of the laser-exposed transistors were tuned to negative values (Fig. 6f). The transfer curves after V_T modulation were shown in Fig. 6g, demonstrating the consistency of the

proposed approach. The V_T of unexposed transistors remained the same after the large-area tuning, as shown in Fig. 6h. The μ_{FE} distribution shows that the mobility increases slightly after UV exposure (Fig. 6i), in agreement with previous results. This demonstration not only shows capability of wafer-scale fabrication and V_T tuning on multiple transistors, but also opens up possibilities for enhancing circuit functionality beyond V_T adjustment.

Fig. 6 | Large-area V_T modulation. a, An image of ultrathin In_2O_3 transistor arrays on a 4-inch wafer. b, Schematics of the automated laser illumination system used for V_T modulation. c, An optical microscope image of UV laser exposure on the designated channel area. d, An optical microscope image of the basic array unit on 4-inch wafer consisted of an 18×25 array (450 transistors). e, A contour plot showing the V_T of the transistors in the array before V_T tuning. f, A contour plot showing the V_T of the transistors in the array after UV exposure. g, The transfer curves of the devices with channel width/length of 10/2 μm in the array after V_T tuning. h, The histogram of V_T of the devices in the array after V_T tuning. i, The histogram of μ_{FE} of the devices in the array after V_T tuning.

2. Why N_2 annealing lead to a negative shift, as show in Fig. 1? What is the difference between $N_2 + O_2$ annealing and UV laser + O_2 annealing?

It is known that metal oxide materials exhibit electron acceptor behavior when O_2 molecules are absorbed on their surface, leading to a decrease in carrier concentration. During annealing in oxygen-deficient environments like N_2 or vacuum, the physically adsorbed O_2 molecules detach, resulting in a negative V_T shift. To investigate the reversibility, we conducted a control experiment by cyclically annealing In_2O_3 devices between N_2 and O_2 environments (Supplementary Fig. 14). The results demonstrate that the V_T shift is reversible, similar to the reversibility observed in UV exposure and O_2 annealing processes. This experiment implies that the mechanisms underlying these two approaches may be identical, with the distinction that UV exposure + O_2 annealing enables local V_T tuning for individual devices, which is unattainable with $N_2 + O_2$ annealing. Moreover, in the back-end-of-line (BEOL) process, reducing the number of annealing steps is preferred to minimize thermal budget and process complexity.

We have revised the main text and added the figure (Supplementary Fig. 14) in the Supplementary Information.

(page 7, line 181)

We conducted cyclically annealing between N_2 and O_2 environments to investigate the mechanism (Supplementary Fig. 14). The results demonstrate that the V_T shift by N_2 and O_2 annealing is reversible, similar to the reversibility observed in UV exposure and O_2 annealing processes. This result implies that the mechanisms underlying these two approaches may be identical.

Supplementary Fig. 14 | Reversibility of V_T tuning in ultrathin In_2O_3 transistors. a, V_T of a 2 nm-thick In_2O_3 transistor during multiple O_2 annealing and N_2 annealing cycles. In_2O_3 transistors are thermally annealed under O_2 for 30 minutes at $150^\circ C$.

3. What is the impact of UV and O₂ annealing on PBS and NBS performance?

The reliability under different bias conditions is a critical issue for practical applications, particularly for oxide-based transistors. To examine the impact of UV and O₂ annealing on device reliability, the PBS and NBS after UV exposure and O₂ annealing were performed (Supplementary Fig. 11). The results show that the In₂O₃ transistors are subjected to the V_T shifts under the gate bias stresses and the direction of the shift depends on the polarity of the biases. The trends observed in PBS and NBS for pristine, UV-exposed, and O₂-annealed devices are similar, suggesting that these processes minimally affect the physical properties of the channel, primarily altering the carrier concentrations.

We added a figure (Supplementary Fig. 11) and the description in the main text and the Supplementary Information.

(page 5, line 132)

Figure 3a shows repeated V_T modulation achieved by multiple O₂ annealing-UV exposure cycles. The mobility of the transistors is maintained at $49 \pm 5 \text{ cm}^2 \text{ V}^{-1} \text{ s}^{-1}$, indicating that the properties of the In₂O₃ transistors are not significantly affected by the tuning V_T processes (Fig. 3b). Additionally, we also conduct the bias stress experiment to investigate behavior and reliability of transistors under different bias conditions after the treatment of O₂ annealing and UV exposure (in Supplementary Fig. S5). A gate voltage of $\pm 15 \text{ V}$ for a total time of 1000 s with both its source and drain electrodes grounded. The transfer characteristics of the device are immediately measured ($<1 \text{ s}$) after bias stress.

SI:

(page 12, line 105)

The results show that the In₂O₃ transistors are subjected to the V_T shifts under the gate bias stresses and the direction of the shift depends on the polarity of the biases. The trends observed in PBS and NBS for pristine, UV-exposed, and O₂-annealed devices are similar, suggesting that these processes minimally affect the physical properties of the channel, primarily altering the carrier concentrations

Supplementary Fig. 11 | PBS and NBS performance with pristine, O₂ annealing and UV exposure. a, Positive bias stress (PBS) and b, Negative bias stress (NBS) of the In₂O₃ transistors. The bias results are performed with devices with thickness of 2 nm and channel width/length of 10/2 μ m.

4 The benchmark in Fig. 4b is not fair. For FE memory or Flash memory, V_t shift is controlled by voltages. However, V_t shift in this work is by UV laser. This is no way to use UV laser to control V_t in memory applications.

We appreciate the reviewer's feedback. We draw comparisons between our method and ferroelectric memory and Flash memory because they all can modulate V_T , despite differing mechanisms and triggers. Our technique is also compared with doping schemes as both involve tuning the carrier concentration, which effectively adjusts the V_T of the channel. In terms of light stimuli for memory applications, we notice that optoelectronic neuromorphic devices have been recently studied intensively [Nature Nanotechnology 14, 776–782, 2019] [Nature 579, 62–66, 2020]. The optoelectronic neuromorphic applications are inspired by the human visual system, showing that memory states can be controlled using light, like lasers, enabling high-speed computations with lower energy consumption. We see potential for our technique in these applications and show an initial demonstration in Supplementary Fig. 18.

To highlight the research regarding the optoelectronic neuromorphic applications, we have added the reference in the manuscript.

(page 9, line 231)

The demonstration of an In₂O₃-based multi-state logic device introduces a new concept to designing reconfigurable multifunctional logic circuits and has the potential for neuromorphic applications (Supplementary Fig. 18)^{40, 41}.

Reviewer #2 (Remarks to the Author):

The manuscript "Wide-range and area-selective threshold voltage

tunability in ultrathin indium oxide transistors" is an interesting report on threshold voltage tunable thin indium oxide transistors. The main noteworthy results shown regard the wide tunability of the threshold voltage in indium oxide TFTs usable for BEOL applications.

The work is of significance to the field of transistors, both flexible and BEOL integrated, also in view of neuromorphic computing applications. It definitively shows advancements as compared to the state of the art of the field, even though the use of thin and very conductive indium oxide layers as semiconductors is opinable. The conclusions are not sufficiently supported, and several major questions are still open (as detailed below).

The methodology used is sufficiently sound, even if more details in the methods are required (see details below).

We thank the reviewer for his/her thorough review of our work. We sincerely appreciate the valuable feedback and positive comments. His/her insights have greatly contributed to enhancing the quality and impact of our study.

The following aspects require a major revision, addressing properly each of them:

1. In the introduction, authors mention an off current (I_{off}) of 20A/mm, which is surely not a current but a current density. Please correct this. Also please clearly talk about I_{on}/I_{off} .

We appreciate the reviewer's feedback. The drain current I_D is reported in units of $\mu\text{A } \mu\text{m}^{-1}$, which represents the current normalized with the "channel width." This normalization allows for consistent comparisons of current values across different channel widths and materials used in electronics. While the term "current" is commonly used, we agree with the reviewer that "current density" is more appropriate.

We have updated the main text accordingly, while maintaining consistency with other papers (*Nature* 593, 211–217, 2021) in the field by using I_D in the figure captions.

(page 3, line 55)

Indium oxide (In_2O_3) recently emerged as a promising channel material for FETs as it can be thinned down to ~1 nm and maintains high electron mobility beyond $100 \text{ cm}^2 \text{ V}^{-1} \text{ s}^{-1}$, showing high on-state drain current density $I_D > 20 \text{ A mm}^{-1}$ (ref. 6,7).

2. Besides desirable and induced V_{th} shifts, authors should clearly discuss and present further V_{th} variations, e.g., with bias stress, light, aging, mechanical forces, etc.

We thank the reviewer for his/her suggestion. We conducted several tests to evaluate the reliability of In_2O_3 transistors, including bias stress, light stress with varying wavelengths and power, and stability in different environments. The characteristics of ultra-thin In_2O_3 transistors under different wavelength illuminations are depicted in Supplementary Fig. 7. The time-dependent V_T variations, reflecting air stability, are presented in Supplementary Fig. 16. Furthermore, we observed V_T variations with different channel lengths, as shown in Supplementary Fig. 13. Additionally, we performed PBS and NBS measurements after subjecting the devices to UV exposure and O_2 annealing to examine their impact on device reliability (Supplementary Fig. 11).

We have added description in the main text and the Supplementary Information accordingly.

3. there are several typos like "atomic layer disposition (ALD)". Please correct them.

We appreciate the reviewer for pointing out the error. We have revised the manuscript, correcting typos and errors, and uploaded the updated version with track changes.

4. Authors should clearly discuss about the disadvantages of using depletion mode TFTs, especially in terms of power consumption and circuit design. Is there any possibility to have a positive V_{th} and how?

It is important to note that the devices demonstrated in this study have the capability to operate in both depletion mode and enhancement mode. We have shown that reducing the channel thickness results in a decrease in carrier concentration, leading to a positive V_T (enhancement mode) for as-fabricated transistors when the In_2O_3 thickness is 2 nm or less. Furthermore, the operational mode of the devices can be modified even after fabrication using the proposed approaches. We show that the implementation of O_2 annealing enables the attainment of positive V_T . This post-fabrication V_T tunability offers increased flexibility for circuit design, as shown in the Fig. 4.

As the reviewer has pointed out, depletion mode TFTs have certain drawbacks, including higher power consumption due to the constant bias voltage requirement for TFT turn-off, complex circuit design necessitating additional circuitry, and reduced noise margin.

To state this clearly, we edited the description in the manuscript:

(page 8, line 212)

After exposing the load transistor to UV light, tuning the selected transistor to depletion mode, the inverter becomes a depletion-load inverter with an enhanced gain of 18 (Fig. 5c). **Note that the depletion load inverter serves as a proof of concept for our proposed technology. Integrating a complementary MOS configuration has the**

potential to further optimize the inverter's performance, resulting in reduced power consumption and improved spatial advantages.

5. Authors only show linear curves. What is the saturation behavior of the TFTs? What higher voltages can be used with such a thin OS?

We have added the I_D - V_D characteristic in the Supplementary Information, which clearly shows the saturation region of the transistor (Supplementary Fig. 6). Regarding the breakdown voltage, we have observed that the gate dielectric (30 nm SiO₂) exhibits high leakage current when the gate voltage exceeds 30 V. Furthermore, initial tests indicate that the device exhibits a critical breakdown electric field exceeding 0.5 MV/cm, primarily attributed to the wide bandgap of the 2 nm In₂O₃. However, evaluating the breakdown voltage of the In₂O₃ requires further refinement of the device structure, which although relevant, is not the primary focus of our current work. We are investigating the breakdown physics and the results will be published in the forthcoming paper.

We have added description in the main text and the Supplementary Information accordingly.

(page 5, line 110)

Figure 1c shows the transfer characteristics (I_D - V_G) of 2 nm In₂O₃ transistors annealed at 200 °C in O₂ and N₂ environments. The transfer characteristics (I_D - V_G) of In₂O₃ below 2 nm is shown in Supplementary Fig. 5 and saturation behavior (I_D - V_D) is shown in Supplementary Fig 6.

SI:

The saturation mobility extracted from $\mu_{sat} = \frac{2I_{D,L}}{WC_{ox}(V_G - V_T)^2}$ at the saturation regime is $60.9 \text{ cm}^2 \text{ V}^{-1} \text{ s}^{-1}$

Supplementary Fig. 6 | I_D - V_D characteristics of an In₂O₃ transistor.

6. Authors should be clearly discussing about the cost and practicability of the proposed technique, also presenting results of possible damages in terms of BEOL laser damage to the other devices/layers?

We appreciate the feedback from all reviewers regarding the practicality of our technique. We fully acknowledge the significance of demonstrating the capability of our approach. To address this concern, we have included Fig. 6 in the manuscript. In this figure, we showcase the fabrication of wafer-scale In_2O_3 transistors on a 4-inch wafer and utilize an industry-level tool (Laser Lift-Off System; K-JET LASER TEK Inc.) for automatic large-area V_T tuning of selected In_2O_3 transistors across the entire wafer.

in terms of cost, it is important to note that indium has been utilized for decades in numerous applications, including displays (e.g., IGZO), transparent electrodes (e.g., ITO), and solar cells (e.g., CIGS), etc. For future BEOL applications, the consumption of indium will be minimal compared to the aforementioned technologies. Further details regarding this aspect can be found in the reviewer's comment No. 10.

The concern of laser damage is examined by employing finite element simulation to assess the temperature variation during laser exposure at different incident power densities. This analysis allows us to evaluate potential risks associated with laser-induced thermal damage.

To discuss the possible damage of laser, we added the description in the manuscript and also the expanded figure (Supplementary Fig. 12) in the Supplementary Materials.

(page 6, line 137)

A gate voltage of ± 15 V for a total time of 1000 s with both its source and drain electrodes grounded. The transfer characteristics of the device are immediately measured (<1 s) after bias stress. Furthermore, to evaluate the potential damage caused by laser exposure, we performed finite element simulations to analyze the temperature changes. The results indicate only a slight increase in temperature at the applied incident power densities (Supplementary Fig. 12).

Supplementary Fig. 12 | Temperature variation with different incident UV laser power density. a, A top view of temperature variation with incident power density of 1×10^2 mW cm⁻². b, A top view of temperature variation with incident power density of 2×10^5 mW. c, A top view of temperature variation with incident power density of 2×10^6 mW. d, A top view of temperature variation with incident power density of 2×10^7 mW. e~h, side view of Supplementary Fig 7a~d. The spot radius of UV laser is 2.54 μ m. The active area of channel is 3 μ m \times 3 μ m of 2 nm In₂O₃ on 30 nm SiO₂ with 1 mm \times 1 mm Si substrate. The results show that the temperature variation is below 10 °C when exposed to a power density of 2×10^2 mW cm⁻² which exceeds the power density applied to the devices by five orders of magnitude.

7. authors present mobility values, are those effective mobility values? how do these values change for different lengths? 8. authors do not show the performance of the devices with different W/L ratios? Actually W/L ratios are not clearly presented in the methods. 14. authors do not show results with different channel dimensions, semiconductors island dimensions. A study of this would be really useful to be added.

We thank the reviewer's comments. We combine the question 7, 8, and 14 and respond here.

For 2 nm thick In₂O₃, the V_T increases with channel length and remains constant with varying channel width. This trend of V_T shift with channel length is the same as the short-channel effect, although the observed V_T shift occurs at channel lengths beyond the typical short-channel limits. Previous reports have observed such V_T shifts in In-based (IGZO) transistors as a function of device geometry, and it was previously attributed to unintentional doping from metal contacts (Appl. Phys. Lett. 102, 083508, 2013). However, our recent findings suggest that the V_T shifts in 2 nm thick In₂O₃ may involve a different mechanism. We have thoroughly investigated this effect, and the paper is currently undergoing revision. The result of different channel active area dimensions has been added at Supplementary Fig. 13. Also, Fig.4a shows the characteristics of different channel thickness.

To state this clearly, we added the following description in the manuscript, add the W/L ratios in the captions of figures, method section and also the expanded figure (Supplementary Fig. 13) in the Supplementary Materials

(page 6, line 142)

The size of the tunable window ΔV_T is dependent on the thickness of the In₂O₃, as shown in Figs. 4a and 4b (V_T shift as a function of channel length is shown Supplementary Fig. 13).

(page 11, line 268)

The active areas of In_2O_3 were defined by lithography with HCl etching with the channel width/length of 10/2 μm .

Supplementary Fig. 13 | Devices characteristics with different channel length (channel width = 10 μm). a. μ_{FE} b. V_{T}

9. The inverters shown are still unipolar and not complementary. Authors should clearly discuss about pro/contra of this method as compared to complementary materials, also in terms of power consumption considering that the threshold voltage is negative.

We appreciate the reviewer's comment. It is important to emphasize that our demonstration of the depletion load serves as a proof of concept, highlighting the viability of our technique in constructing a logic circuit. While the complementary MOS (CMOS) configuration does offer improved performance compared to unipolar MOS, it is worth noting that CMOS also requires the tuning of the V_{T} to optimize power efficiency. At present, because high-performance OS p-type transistors are not readily available, we choose to show the unipolar inverter as a suitable alternative in our study.

To state this clearly, we add the description in the manuscript:

(page 8, line 212)

After exposing the load transistor to UV light, tuning the selected transistor to depletion mode, the inverter becomes a depletion-load inverter with an enhanced gain of 18 (Fig. 5c). Note that the depletion load inverter serves as a proof of concept for our proposed technology. Integrating a complementary MOS configuration has the potential to further optimize the inverter's performance, resulting in reduced power consumption and improved spatial advantages.

10. Authors should discuss about availability of indium, when presenting the different semiconductor materials. A true comparison with all relevant parameters is required. Indium has been widely utilized for several decades, leading to the establishment of a well-developed supply chain and infrastructure. This availability has made indium a practical choice for numerous applications, including displays (e.g., IGZO), transparent electrodes (e.g., ITO), and solar cells (e.g., CIGS), etc.

Regarding future back-end-of-line (BEOL) applications, it is worth noting that the thickness of In_2O_3 transistors will be only a few nanometers. Consequently, we anticipate that the consumption of indium will be low compared to the aforementioned technologies.

To provide detailed information on this aspect, we have included a supplementary table (Table S1) and a description in the manuscript.

(page 4, line 88)

Indium has been utilized across a variety of applications (element scarcity is shown in Table S1), including displays (e.g., IGZO), transparent electrodes (e.g., ITO), and solar cells (e.g., CIGS). In this study, ultrathin In_2O_3 is employed as the channel material of a transistor, as shown schematically in Figure 1a.

Element	%	Price (USD/kg)
O	46.1	0.154
Si	28.2	1.7
Al	8.23	1.79
S	0.035	0.093
C	0.02	0.122
Zn	0.007	2.55
Cu	0.006	6
N	0.0019	0.14
Ga	0.0019	148
Sn	0.00023	18.7
As	0.00018	0.999-1.31
Ge	0.00015	914 - 1010
Mo	0.00012	40.1
In	0.000025	167
Sb	0.00002	5.79
Se	0.000005	21.4
Te	1.00E-07	63.5

Table S1 | The abundance and price of common semiconductor element.

11. Authors use ALD deposited indium oxide? could also sputtering (mostly used in OS processing) be used?

It is possible to use sputter technique to deposit In_2O_3 thin film. However, the roughness could be an issue and it could severely affect the performance of the device especially when the thickness of the channel is in the same range as the film roughness. ALD provides numerous advantages over sputtering because it has more precise control of film thickness with high film quality and uniformity (AFM data is shown in Supplementary Fig. 2). It could also provide conformal coating capabilities on curved or trench structures with excellent step coverage on high aspect ratio features. In addition, the ALD deposition is performed at low deposition temperatures, which is particularly advantageous for BEOL applications.

12. Could ALD be used even below 2 nm?

ALD deposition could achieve In_2O_3 below 2 nm and the mobility of In_2O_3 transistors decreases while the thickness scale below 1 nm. We have added a figure in the supplementary Information (Supplementary Fig. 5)

(page 5, line 110)

Figure 1c shows the transfer characteristics (I_D - V_G) of 2 nm In_2O_3 transistors annealed at 200 °C in O_2 and N_2 environments. The transfer characteristics (I_D - V_G) of In_2O_3 below 2 nm is shown in Supplementary Fig. 5 and saturation behavior (I_D - V_D) is shown in Supplementary Fig. 6.

Supplementary Fig. 5 | Devices characteristics with 1/1.5 nm channel thickness and channel width/length of 10/2 μm .

13 authors do not really comment on the semiconductivity of such a thin indium oxide layer. what about the saturation mobility?

The semiconductor properties of such a thin indium oxide were previously explained by the trap neutral level (TNL) model, which suggests that the Fermi level aligns within the bandgap of In_2O_3 as the thickness reduces to the quantum confinement regime [Nano Lett. 21, 500–506, 2021]. Supplementary Fig. 15 illustrates the extracted bandgap from the scanning tunneling spectroscopy (STS) measurement, providing further evidence of the semiconducting nature of the atomically thin In_2O_3 films. The saturation mobility extracted from $\mu_{sat} = \frac{2I_{DL}}{WC_{OX}(V_G - V_T)^2}$ at the saturation regime is $60.9 \text{ cm}^2 \text{ V}^{-1} \text{ s}^{-1}$

15. Authors conclude that this method is comparable to doping or capacitively charged techniques. it is indeed not clear in what this method is better, this is in fact very important.

The proposed methods achieve a similar effect as chemical doping, but with the advantage of reversibility and area selectivity. Unlike chemical doping in 2D materials, which poses significant challenges, our method demonstrates the ease of achieving effective carrier concentration tuning in ultrathin In_2O_3 . Additionally, our approach minimizes the number of fabrication processes, enhancing its potential for successful integration in BEOL applications. To clarify the difference, we edited the description in the manuscript:

(page 6, line 160)

The tunable ranges of n_{2D} achieved in In_2O_3 transistors vary from 10^{10} cm^{-2} to 10^{13} cm^{-2} depending on the thickness. 2 nm In_2O_3 exhibits the widest n_{2D} tunable window ($2 \times 10^{10} \text{ cm}^{-2}$ to $2 \times 10^{12} \text{ cm}^{-2}$), indicating the proposed method is competitive among various 2D doping techniques. The proposed methods offer a comparable effect to chemical doping, while providing the advantages of reversibility and area selectivity. In contrast to the substantial challenges associated with chemical doping in 2D materials, our method showcases the ease of achieving effective carrier concentration tuning in ultrathin In_2O_3 . Moreover, our approach minimizes fabrication processes, thereby enhancing its potential for seamless integration in BEOL applications.

Reviewer #3 (Remarks to the Author):

In this report, Tseng et. al. use a combination of thermal annealing and UV light exposure to tune the threshold voltage of transistors based on very thin (< 5 nm) films of In_2O_3 . These are interesting devices which are gaining more interest in the community, and the authors contribution provides useful insights into how one might be able to tune the threshold voltage of these transistors. Unfortunately, however, I cannot recommend this manuscript for publication in its current form. While the results are reasonable, they primarily describe a strategy to control devices in the group's own lab, rather than being of use to the general audience. To elaborate on this slightly, these are results for a specific device design, no device-to-device variation is provided, and the effect is parameterised in terms of voltage, which depends on the properties of the interface, device dimensions and so on. Overall, I don't disagree with the methods, motivation, nor the conclusions drawn; I just feel the contribution is not broad enough to warrant publication in Nature Communications unfortunately. To be able to recommend this paper for publication in Nature Communications I would need to see:

We thank the reviewer's feedback about the applicability of our research to a broader audience. We appreciate the opportunity to address this point. We have added Fig. 6, highlighting the potential integration of our technique with industrial-level equipment. This demonstrates the scalability of our approach for wafer-scale implementation. Specifically, we achieve large-area V_T modulation in an 18×25 FET array on a 4-inch wafer, emphasizing the practical and industrial relevance of our research.

1. Quantification of device-to-device variation with multiple, identically fabricated, devices on different substrates.

We fabricate 10 separate batches of devices using an independent and identical process to examine the device-to-device variation. For each batch, we measured 5 devices and gathered a total of 50 independent data. The results are presented in Supplementary Fig. 3a-c show the extracted V_T and field effect mobility, respectively, indicating minimal device-to-device variation in this manufacturing process.

We have added a figure in the supplementary Information (Supplementary Fig. 3)

Supplementary Fig. 3 | Devices sampling. a, Transfer curves b, Histogram of V_T c, Histogram of μ_{FE} .

(page 4, line 98)

The high-resolution transmission electron microscopy (HRTEM) image demonstrates the amorphous nature and atomic-level uniformity of the ALD In_2O_3 films, which effectively accounts for the minimal device-to-device variation (Fig. S8).

2. A description of how V_T depends on geometry. I am confident that it would depend on both W and L .

The reviewer's anticipation is correct. For 2 nm thick In_2O_3 , the V_T increases with channel length and remains constant with varying channel width. This trend of V_T shift with channel length is the same as the short-channel effect, although the observed V_T shift occurs at channel lengths beyond the typical short-channel limits. Previous reports have observed such V_T shifts in In-based (IGZO) transistors as a function of device geometry, and it was previously attributed to unintentional doping from metal contacts (Appl. Phys. Lett. 102,

083508, 2013). However, our recent findings suggest that the V_T shifts in 2 nm thick In_2O_3 may involve a different mechanism. We have thoroughly investigated this effect, and the paper is currently undergoing revision.

To state this clearly, we added the following description in the manuscript and add the W/L ratios in the captions of figures and also the expanded figure (Supplementary Fig. 13) in the Supplementary Materials

(page 6, line 141)

The size of the tunable window ΔV_T is dependent on the thickness of the In_2O_3 , as shown in Figs. 4a and 4b (V_T shift as a function of channel length is shown Supplementary Fig. 13).

Supplementary Fig. 13 | Devices characteristics with different channel length (channel width = 10 μm). a. μ_{FE} b. V_T

3. Confirmation that this is not unique to the exact structure depicted in Figure 1a. It would not be necessary to consider every possibility of course. But it would need to be confirmed that the V_T shifts were in the same directions if using, say, different electrode materials, for example.

To verify the reproducibility, we tested devices with identical fabrication processes but different metal contacts (Pd, Pt) and substrates (10 nm HfO_2 as the back gate dielectric). All results exhibited consistent trends. The device shows negative V_T shift after UV exposure and positive V_T shift after O_2 annealing, regardless of substrate and metal contacts.

To state this clearly, we added the following description in the manuscript and add expanded figure (Supplementary Fig. 10) in the Supplementary Materials

(page 5, line 123)

Note that UV exposure causes an accumulation of V_T shift, leading to a saturation point of -15 V with a tuning resolution of 0.05 V within a tunable window ΔV_T of 21 V (as shown in Supplementary Fig. 5). To verify reproducibility of our approach, we tested devices with identical fabrication processes but different metal contacts (Pd, Pt) and substrates (10 nm HfO₂ as the gate dielectric). All results exhibited consistent trends, regardless of substrate and metal contacts. (Supplementary Fig. 10).

Supplementary Fig. 10 | Transfer curves for different substrate and electrodes with O₂ annealing and UV exposure. a, characteristics of 2 nm In₂O₃ on ALD 10 nm HfO₂ devices with 40 nm Ni electrodes. b, characteristics of 2 nm In₂O₃ on ALD 30 nm SiO₂ devices with 40 nm Pt electrodes. c, characteristics of 2 nm In₂O₃ on ALD 30 nm SiO₂ devices with 40 nm Pd electrodes.

4. There needs to be some comment on the stability / longevity of this strategy. E.g. if UV light is used to shift the threshold voltage, how long does the state remain for? I suspect these V_T shifts are quasi-stable.

Retention measurements were performed to assess the stability of the V_T tuning strategy proposed in this study, as shown in Supplementary Fig. 16. The results demonstrate that the V_T gradually reverts back to the value of the pristine sample when exposed to air, while maintaining a stable V_T when placed in a high vacuum environment. This sensitivity to the ambient environment can be attributed to the inherent characteristics of oxide semiconductors, which are susceptible to defects associated with oxygen vacancies. These findings suggest that stability can be enhanced by effectively isolating or passivating the channel to minimize its exposure to the atmosphere.

We have revised the manuscript accordingly to state this clearly.

(page 8, line 198)

Additionally, the V_T exhibits almost no change over time in an ultra-high vacuum environment (10^{-9} torr), which further highlights the crucial role of oxygen adatoms in this process. The results suggest that the stability could be improved by implementing effective isolation or passivation to minimize the exposure of In_2O_3 to the atmosphere.

5. The introduction reads as if In_2O_3 is competing with silicon. In_2O_3 is better suited for large-area applications like display backplanes and so-on than for information processing. I would consider re-writing this.

We thank the reviewer's suggestion. Indeed, oxide-based semiconductors, e.g., IGZO, are already commercialized and seen in entertainment products such as displays. Recent progresses show that In_2O_3 transistors can achieve high mobility at thicknesses below 2 nm, while remaining compatible with existing processes (i.e., growth temperature below BEOL thermal budget and ALD feasibility). This new finding suggests their device scaling capability to meet current technology nodes, particularly, enabling their applicability in back-end-of-line (BEOL) processes for monolithic 3D integration. This discovery has expanded beyond their traditional use in displays. Consequently, an increasing number of papers related to the new findings have been published in prestigious electronic conferences such as IEDM and VLSI in the past year. Rather than competing with silicon-based CMOS, oxide semiconductors are expected to serve as a complementary technology, playing a role in BEOL integration.

To emphasize the current research trends in this area, we have revised the manuscript accordingly.

(page 3, line 55)

Indium oxide (In_2O_3) recently emerged as a promising channel material for FETs as it can be thinned down to ~ 1 nm and maintains high electron mobility beyond $100 \text{ cm}^2 \text{ V}^{-1} \text{ s}^{-1}$, showing high on-state current $I_{\text{ON}} > 20 \text{ A/mm}^{6,7}$. This advancement allows for the scaling of In_2O_3 to align with modern technology nodes, and is expected to complement silicon-based systems for future back-end-of-line (BEOL) integration, expanding its range of applications beyond display technology.

6 Can the authors comment on the continuity / roughness of the films. 2 nm is very thin, and with ALD I would not be surprised if there were pinholes or significant thickness variability.

The film is quite uniform as examined by the AFM ($R_q = 0.195$ nm and $R_a = 0.146$ nm). The AFM data is added and shown in Supplementary Fig. 2. We want to emphasize that in situations where the material thickness is only a few nanometers, a significant portion of the surface roughness is determined by the quality of the underlying substrate, and the substrate we used (30 nm $\text{SiO}_2/\text{P}^{++}\text{Si}$) to grow ALD In_2O_3 has similar uniformity.

To state this clearly, we added edited the description in the manuscript and the expanded figure (Supplementary Fig. 2) in the Supplementary Materials

(page 4, line 98)

The high-resolution transmission electron microscopy (HRTEM; Supplementary Fig. 1) image reveals the amorphous nature and the atomic level uniformity of the In_2O_3 films is confirmed by AFM (Supplementary Fig. 2), which effectively accounts for the minimal device-to-device variation (Supplementary Fig. 3).

Supplementary Fig. 2 | AFM data with 2 nm In_2O_3 . The roughness of $R_q = 0.19$ nm and $R_a = 0.14$ nm.

7. The following statement needs to be rephrased slightly. “This is because the electronic property of OS is insensitive to crystallinity. Their transport is determined by the overlapping of s orbitals between neighboring metal atoms (e.g. indium) and is not affected by structural disorders.” The effect of crystallinity on transport is certainly lower in metal oxides compared to covalent systems (e.g. a:Si-H) but order does still have some affect. I would use the phrase “less affected by structural disorders” rather than “not affected by structural disorders”.

We thank the reviewer for the suggestion. We have edited the description in the manuscript:

(page 5, line 103)

Their transport is determined by the overlapping of s orbitals between neighboring metal atoms (e.g. indium) and is less affected by structural disorders.

8. For the heatmaps (e.g. Figure 2(c), Figure S4), how many data points are used? This looks like it is interpolated / smoothed? 9. For Figure 2(c) (and similar measurements), what was the time between measurements and what were the conditions between measurements? I strongly suspect that illumination and annealing history would have a notable impact on these results.

We thank the reviewer's comments. We combine the question 8, 9 and respond here.

For the heatmaps (Figure 2(c), Supplementary Fig. 8), 70 data points (7 set of different power density and 10 set of time interval) were used. Because the V_T of the device is highly reversible between UV exposure and O_2 (Fig. 3a), we are to reset the V_T before we perform the UV exposure measurement.

To state this clearly, we added the description in the caption of Figure 2(c) and Supplementary Fig. 8.

Fig. 2 | V_T tuning in ultrathin In_2O_3 transistors via UV exposure. a, A schematic of V_T tuning in ultrathin In_2O_3 transistors through UV exposure combined with thermal annealing. b, Transfer curves of ultrathin In_2O_3 transistors with channel width/length of 10/2 μm after exposure to UV light. after exposure to UV light. c, A contour plot of V_T shifts as a function of UV exposure time and power density. Devices were annealed at 150 $^{\circ}C$ in O_2 for 30 minutes to reset the V_T before each UV exposure measurement. The absorbed power density increasing from 1×10^6

mW cm⁻² to 1 × 10⁸mW cm⁻² for exposures times from 30 s to 300 s under 365 nm laser illumination. The measurement time interval is 30s and the transfer characteristics of the device are immediately measured (<5s) after UV illumination. The plot consists of 70 data points. (7 set of different power density and 10 set of time interval)

Supplementary Fig. 8 | A contour plot of threshold voltage variation with absorbed power density increasing from 0.1 mW cm⁻² to 10 mW cm⁻² for exposures times from 10 s to 100 s under 532 nm laser illumination. Devices were annealed at 150 °C in O₂ for 30 minutes to reset the V_T before each UV exposure measurement. The measurement time interval is 10s and the transfer characteristics of the device are immediately measured (<5s) after 532 laser illumination. The plot consists of 70 data points. (7 set of different power density and 10 set of time interval)

10. I'm not sure Figure 2a really adds anything. It feels more like a Table of Contents graphic than a subfigure.

We appreciate the reviewer's comment regarding Figure 2a. We believe it serves a key purpose in demonstrating how illumination can spatially tune individual devices. The inset also provide information about the device's appearance under an optical microscope. Therefore, we would like to retain it.

11. I think the units for mobility on page 5 should be cm²/Vs not m²/Vs.

Yes, we have corrected the error. Thank you.

REVIEWERS' COMMENTS

Reviewer #1 (Remarks to the Author):

I still cannot recommend publishing this paper for the following reasons.

1. Vt tuning is not a critical challenge for oxide semiconductor. For example, to increase thickness of In₂O₃ can easily shift Vt negatively. The method developed in this work doesn't have a broad interest and no obvious advantage over other approaches.
2. The method developed in this work doesn't demonstrate a high performance. For example, after UV exposing, the variation becomes huge.
3. Although the authors show the laser spot can move by controlling the stage, however, it seems still impossible to form "wafer scale uniform" and fast tuning of Vt by such method. So large-area fabrication is not likely to be possible.

Reviewer #2 (Remarks to the Author):

The manuscript "Wide-range and area-selective threshold voltage tunability in ultrathin indium oxide transistors" has been revised according to the first review comments.

Reviewer #3 (Remarks to the Author):

Overall, I am satisfied by the revisions made in response to my comments and am now happy to recommend this manuscript to be published in Nature Communications.

Answers to Reviewers' Comments

Reviewer #1 (Remarks to the Author):

I still cannot recommend publishing this paper for the following reasons.

1. V_t tuning is not a critical challenge for oxide semiconductor. For example, to increase thickness of In_2O_3 can easily shift V_t negatively. The method developed in this work doesn't have a broad interest and no obvious advantage over other approaches.

Although V_T can be controlled by adjusting the thickness of In_2O_3 discusses (as also discussed in this study), the thickness is predetermined during fabrication, making post-fabrication adjustments to V_T through thickness impractical. Our proposed method introduces a complementary approach for reversible V_T tuning that can be implemented after fabrication. We believe this adequately underlines the significance of our work.

2. The method developed in this work doesn't demonstrate a high performance. For example, after UV exposing, the variation becomes huge. So large-area fabrication is not likely to be possible.

Please see the next response.

3. Although the authors show the laser spot can move by controlling the stage, however, it seems still impossible to form "wafer scale uniform" and fast tuning of V_t by such method.

We appreciate the concern raised regarding the 'wafer scale uniformity' of our method, particularly the device-to-device variation observed during wafer-scale V_T tuning. This study primarily focuses on demonstrating the feasibility of our proposed approach, therefore we did not extensively optimize the wafer-scale transistor fabrication processes or V_T tuning parameters for the wafer-level demonstration. In the future, the device-to-device variation, induced by the variation in film quality across the wafer, may potentially be mitigated through the utilization of industrial-level ALD tools. Furthermore, the V_T tuning precision could be improved by the incorporation of patterned photomasks. We have revised the manuscript with further discussion on these future directions:

(page 9; line 245)

Note that the device-to-device variation, induced by the variation in film quality

across the wafer, may potentially be mitigated through the utilization of industrial-

level ALD tools. Furthermore, the V_T tuning precision could be improved by the incorporation of patterned photomasks.

Reviewer #2 (Remarks to the Author):

The manuscript "Wide-range and area-selective threshold voltage tunability in ultrathin indium oxide transistors" has been revised according to the first review comments.

Reviewer #3 (Remarks to the Author):

Overall, I am satisfied by the revisions made in response to my comments and am now happy to recommend this manuscript to be published in Nature Communications.